# Chromatin Structure and Dynamics: Focus on Neuronal Differentiation and Pathological Implication

**DOI:** 10.3390/genes13040639

**Published:** 2022-04-02

**Authors:** Sophie A. Nothof, Frédérique Magdinier, Julien Van-Gils

**Affiliations:** 1Marseille Medical Genetics, Aix Marseille University, Inserm, CEDEX 05, 13385 Marseille, France; sophie.nothof@etu.univ-amu.fr (S.A.N.); frederique.magdinier@univ-amu.fr (F.M.); 2Reference Center AD SOOR, AnDDI-RARE, Inserm U 1211, Medical Genetics Department, Bordeaux University, Center Hospitalier Universitaire de Bordeaux, 33076 Bordeaux, France

**Keywords:** neuronal differentiation, histone modification, epigenetics, chromatin, acetylation, methylation, chromatinopathies

## Abstract

Chromatin structure is an essential regulator of gene expression. Its state of compaction contributes to the regulation of genetic programs, in particular during differentiation. Epigenetic processes, which include post-translational modifications of histones, DNA methylation and implication of non-coding RNA, are powerful regulators of gene expression. Neurogenesis and neuronal differentiation are spatio-temporally regulated events that allow the formation of the central nervous system components. Here, we review the chromatin structure and post-translational histone modifications associated with neuronal differentiation. Studying the impact of histone modifications on neuronal differentiation improves our understanding of the pathophysiological mechanisms of chromatinopathies and opens up new therapeutic avenues. In addition, we will discuss techniques for the analysis of histone modifications on a genome-wide scale and the pathologies associated with the dysregulation of the epigenetic machinery.

## 1. Introduction

Unlike prokaryotic organisms, eukaryotic DNA needs to be compacted to facilitate its localization within the cell nucleus. The level of DNA and chromatin compaction is defined as two different states depending on its permissiveness to the transcriptional machinery. Euchromatin is defined as an “open” permissive state of chromatin and heterochromatin as the “closed”, non-permissive state [1]. The open or closed chromatin states are modulated by post-translational modifications (PTM) of histones that participate in epigenetic regulation processes. First evoked in 1942, epigenetics defines the processes leading to the modifications of gene expression in the absence of changes in the DNA sequence. Epigenetic processes include DNA methylation, post-translational histone modifications and implicate non-coding RNAs that modulate chromatin accessibility [2].

In the 60s, PTMs of the amino-terminal tails of histones have been identified as the targets of different epigenetic factors dubbed writers, readers and erasers. The epigenetic writers affix epigenetic marks on histones amino acids. Epigenetic erasers remove deposited PTMs, and readers are able recognize specific epigenetic marks and recruit chromatin remodeling complexes able to translate the associated information. Genetic defects in genes encoding a number of epigenetic players lead to pathologies defined as chromatinopathies such as Rubinstein Taybi syndrome (RSTS; OMIM #180849, OMIM #613684), Kabuki syndrome 1 (KS1; OMIM #147920) or SOTOS syndrome (OMIM #277590) among others [3] or are observed in cancers [4].

Post-translational modifications of histones play a key role in gene regulation, in particular during neurogenesis and differentiation of the three neural lineages. Neurogenesis occurs during embryogenesis but also in the adult brain [5]. Methylation and acetylation are the most studied histone PTMs. These marks are involved in neurogenesis. Their in vitro and in vivo modulation impact neuronal differentiation [6,7].

In this review, we will discuss the chromatin structure and dynamics, focusing on post-translational modification of histones during neuronal differentiation and the consequences of defects in the epigenetic machinery in pathologies. Finally, we will discuss the methods of analysis of these large-scale histone modifications.

## 2. Chromatin Structure

Human genetic information, which is supported by DNA, consists of 3 × 10^9^ base pairs (bp), i.e., 6 × 10^9^ bp in the nucleus of a diploid cell [8]. This corresponds to approximately 2 m of DNA, contained within a nucleus of 5 to 10 µm in diameter [9]. DNA is condensed by association to small basic proteins, the histones, allowing a compaction level of 10,000 to 20,000 times (Figure 1) [10].

### 2.1. Heterochromatin and Euchromatin

The different states of chromatin play essential roles in all DNA transactions, i.e., transcription, repair and replication of DNA [11]. Two states of compaction are defined. Heterochromatin first described by Emil Heitz to distinguish chromosomal regions that remained strongly stained throughout the cell cycle from those lightly stained with acetocarmine Heterochromatin is transcriptionally repressed while euchromatin is permissive to gene transcription [12].

Two forms of heterochromatin exist: constitutive and facultative heterochromatin. Facultative heterochromatin can switch to euchromatin, permissive to the transcriptional machinery. The repressive H3K27me3 mark regulated by the Polycomb repressive complexes is enriched at facultative heterochromatin [13]. Constitutive heterochromatin is associated with a low level of acetylation, CpG methylation, and is enriched in histone repressive marks such as H3K9me3. Heterochromatin proteins 1 (HP1) bind to the repressive H3K9me3 marks and are involved in the formation and propagation of heterochromatin [14]. Constitutive heterochromatin is mainly clustered in regions enriched in repetitive DNA sequences (centromere and pericentromeres in particular) and gene-poor regions [13].

Moreover, heterochromatin is preferentially located at the nuclear periphery in interactions with the nuclear envelope. In the 3D hierarchy of the genome, two compartments have been defined: the A compartment (euchromatin) enriched in H3K27ac and H3K39me3 and transcriptionally active, and the B compartment (heterochromatin), associated with a silent state of transcription, enriched in H3K9me3 [15]. In contrast to this configuration found in most eukaryotic nuclei, some nuclei display a so-called inverted compartmentalization with a dense nuclear center surrounded by a more decondensed region [16]. This type of nuclear architecture is found, for instance, in the rod photoreceptors involved in the black and white vision in mammals.

In addition to these two conventional compartments, a compartment I has been described in colon cells. In the normal colon, this I compartment is located at an intermediate position between the two canonical A and B compartments, interacts with these two compartments and is associated with a relatively low level of transcription. It is distinguished from the two other compartments by an enrichment in H3K27me3 [15]. The study of cells with inverted nuclei (rod photoreceptors, lamin B receptor-null thymocytes) and conventional nuclei (non-rod retinal neurons, wild type thymocytes from mice) using polymer simulations to reconcile microscopy and Hi-C data revealed that interactions between heterochromatic regions are essential in the A and B compartmentalization of inverted nuclei but also of conventional type nuclei.

The living chromatin model, which allows the study of coupling between chromatin folding and epigenetic regulation, shows that the maintenance of the epigenome is regulated by 3D compartmentalization. This compartmentalization allows the local concentration of epigenome effectors, increasing their capacity to diffuse an epigenomic signal at a long distance [17]. In addition, lamina-heterochromatin interactions are required to establish the conventional nuclear architecture [18].

### 2.2. DNA Compaction

The first fundamental level of higher-order chromatin structure is called the nucleosome. This structure is composed of an octamer of histones and DNA wrapped around it [19,20,21]. The nucleosome consists of two dimers of H2A-H2B and a tetramer H3-H4 that allow the left winding, in a helical way, of 145 to 147 bp of DNA [22,23]. This structure forms the nucleosome core particle and has a molecular weight of approximately 205 kDa. The nucleosome also affects the pitch of the helix; the DNA alone, type B, has 10.5 bp/turn of the helix; the DNA wrapped around the histones undergoes a slight twist leading to a helix pitch of 10.2 bp/turn [22,24].

The nucleosome core particle (NCP) has an axis of symmetry. The axis is in a plane perpendicular to the H2A/H2B dimer and passes through the H3/H4 dimer. The H2A/H2B dimer contains six amino acids of H2A (Glu56, Glu61, Glu64, Asp90, Glu91, Glu92) and two amino acids of H2B (Glu105, Glu113) that create a negatively charged environment. This acid patch is involved in chromatin compaction by allowing interactions with the N-terminal tail of the adjacent histone H4 nucleosome [25,26].

In addition to the nucleosome core particle, the linker histone or internucleosomal histone H1 forms a new chromatin structure called chromatosome [27]. H1 has a molecular weight of 21 kDa and promotes interaction between adjacent nucleosomes. The linker histone is composed of approximately 200 amino acids and binds to about 10 to 60 bp of DNA (called linker DNA). Electrostatic interactions between lysine or arginine residues of chromatosome histones and DNA phosphate groups stabilize the nucleosome [10,28,29,30].

The second level of DNA compaction involves the formation of the 30 nm chromatin fiber [31]. H1 plays an essential role in its formation since it cannot be formed in vitro if linker histones are depleted. Interactions between the acid patch and the tail of histone H4 are also involved in its formation. The structure of the 30 nm fiber is still debated, and two main models are proposed: the solenoid model and the zigzag model. In the zigzag model, odd nucleosomes are stacked on top of each other as are the even nucleosomes. Interactions involve nucleosome N and N + 2. In the solenoid model, the interactions are made between the N and N + 1 nucleosomes, which allow the nucleosomes to follow one after the other [31,32].

At the 3D level, chromatin fiber can form loops through local interactions that delimitates topologically associated domains (TADs). DNA sequences inside a TAD interact with each other more frequently than sequences outside. These TADs are typically delimited by the interaction of the anchor CTCF protein and cohesin. These regions delineate enhancer/promoter interactions and thus play a role in gene expression [33]. The CTCF protein also participates in the stability of adjacent antagonistic epigenomic domains [17]. TADs can be found in the three genomic compartments A, B or I and interact preferentially with each other within the same compartment [15].

### 2.3. Histone Structure and Variants

Histones, like other proteins, are synthesized in the cytoplasm. To prevent histone mismatch, so-called histone chaperone proteins bind to neosynthesized histones involved in the transport of histones to the nucleus [34]. These chaperones also prevent DNA-non-specific bonds and histone degradation. ATP-dependent chromatin remodeling complexes are associated to these histone chaperones and required for nucleosome positioning. These complexes are involved in the incorporation of histone variants and influence nucleosome spacing, sliding or removal. Four families of chromatin remodelers have been defined and classified according to the sequence and structure of the ATPase domain: SWI/SNF, ISWI, CHD and INO80 [35].

Histones from the nucleosome core particle comprise a secondary structure known as the histone fold, amino- and carboxyterminal extensions, an amino-terminal tail and a carboxyterminal tail for H2A. The histone fold is constituted by three helixes connected by loops according to a α1-L1-α2-L2-α3 model. The α1 and α3 helix are relatively short (9 to 14 amino acids), unlike the α2 helix with an average of 29 amino acids [22,23]. The secondary structure of the histone fold is maintained between the four nucleosome core particle histones despite low sequence retention. The histone fold promotes protein–protein interactions that will be used for heterodimerization of H2A/H2B and H3/H4 [36]. The N-terminal ends do not have a defined secondary structure. The linker histone is composed of unstructured amino- and carboxyterminal tails and an apolar central globular area. The C-terminal tail is basic [37].

In addition to the conventional canonical histones, several histone variants, defined as non-allelic isoforms of canonical histones exist [38]. Variants have specific characteristics that modify the nucleosome structure with specific functions [39]. Histones H2A and H3 have the higher number of variants. Among them, CENP-A, a histone H3 variant specific to centromeres, plays a key role in the assembly of the kinetochore complex during mitosis [40]. The γ-H2AX variant, phosphorylated on serine 139, by the ATR and ATM kinases of the cell cycle checkpoint is recognized by DNA repair effectors and involved in DNA damage response [41]. MacroH2A accumulates at double strand DNA breaks. The macroH2A1.1 splice variant interacts with the KDM5A lysine demethylase, which is recruited to DNA damage sites. Subsequently, H3K9me3 demethylation promotes the recruitment of the ZMYND8-NuRD complex. This complex causes transcriptional repression and repair of double-strand breaks by homologous recombination [42,43,44].

## 3. Histone Post-Translational Modifications

In the 1960s, it was discovered that histones from the nucleosome core particle may be subject to PTM of amino acids located mainly in their amino-terminal tail [45] but also at the carboxy-terminal tail for histone H2A [26]. Histone tails are unstructured, flexible domains, enriched with lysine and arginine amino acid, and their strategic position outside the nucleosome core allows easy access to PTM catalyzing enzymes and other proteins [22]. These histone tails represent 25% of the total mass of histones [46]. More than 60 histone residues have been identified as potentially modified post-translation (Figure 2) [47].

Histone modifications affect chromatin compaction, thus regulating gene transcription and are therefore associated with epigenetic processes. Enzymes that catalyze PTMs are considered as epigenetic writers; so-called readers are proteins able to read these PTMs, while erasers remove them. Histones undergo different types of PTMs, such as acetylation, methylation, phosphorylation, ubiquination, polyADP-ribosylation, SUMOylation, deimination, citrullination, proprionylation, crotonylation, or isomerization of proline residues [47,48,49,50,51,52,53,54]. These PTMs form a “histone code” that affects the higher chromatin structure, inter-histone interactions, DNA-histone interactions and the recruitment of non-histone protein [53]. Acetylation and methylation, the most studied PTMs for their role in transcriptional regulation are detailed below. Post-translational modifications in the same histone or between adjacent histones may interact in an antagonistic or synergistic manner to modulate cellular processes [55,56].

### 3.1. Histone Acetylation and Deacetylation

The addition of an acetyl group on histones was discovered in 1964 [45]. The reaction is catalyzed by histones acetyl transferase (HAT) using acetyl CoA as a cofactor allowing the establishment of a covalent bond between an acetyl group and the lateral chain ε-amino of lysine. At a physiological pH, electrostatic interactions are formed between histone proteins (positively charged by lysine and arginine) and DNA (negatively charged by phosphate groups), allowing DNA compaction. The loss of the positive charge of lysine weakens the electrostatic bonds, leading to a relaxation of the chromatin, which favors the access of transcription factors to *cis*-regulatory sequences (Figure 3) [57].

Since HATs do not exclusively target histones, a more generic name, K-acetyltransferases (KAT) has been given [58]. Six families are distinguished in this KAT superfamily: the Gcn5-related N-acetyltransferase (GNAT) family, the MYST family, the p300/CBP family, the HAT family related to transcription factors, the cytoplasmic KAT family and the nuclear receptor co-activators family (Table 1) [58,59]. Acetylated proteins can be recognized, for example, by bromodomains, Yeats domains and double PHD finger [60]. Acetylation, like other histone PTMs, is a reversible modification. Histone deacetylase (HDAC) catalyze the removal of acetyl groups on histones, generally assimilated to transcriptional repression (Figure 3). These enzymes have a low substrate specificity allowing them to target several histone residues [29]. The eighteen HDACs are classified into four classes according to their sequence and mechanism of action. Class I HDACs, which include HDAC1 to 3 and HDAC8, are nuclear proteins. HDAC4 to 7, HDAC9 and HDAC10 belong to Class II. Class III are NAD-dependent and include members of the Sirtuin subfamily (SIRT 1 to 7). HDAC11 is the only member of class IV. Class I, II and IV are zinc-dependent [60].

### 3.2. Methylation and Demethylation of Histones

Another important type of histone PTMs is methylation (CH3) of lysine (K) and arginine (R) or histidine (H) residues [61]. This reversible reaction is catalyzed by K-methyltransferase (KMT) and arginine methyltransferase protein (PRMT) [61,62,63]. These enzymes act on both chromatosome histones, histones not yet incorporated into chromatin and non-histone proteins. The addition of a methyl group does not affect the global histone charge. Lateral chains of lysine and arginine can be targeted by methylation. Lysine can be mono-, di- or trimethylated and arginine can be mono- or dimethylated. Arginine dimethylation can be symmetrical (noted me2s) or asymmetrical (noted me2a). S-adenosyl-L-methionine is the donor of methyl groups that is added to the ε-amino group of the target amino acid. Methylation is involved in the condensation or relaxation of chromatin depending on its position on histones [64] (Figure 4).

KMT can be subdivided into two groups: KMT containing the Su(var)3–9 domain, Enhancer of Zeste, trithorax (SET, for example SUV39H1 and SUV39H2) and lysine methyltransferases, which do not contain a SET domain (e.g., Disruptor of Telomeric silencing 1 Like or DOT1L). The SET domain is involved in the catalytic activity at the histone tail, while the second group targets the central part of the histone. Methylation readers are composed of different types of domains allowing them to associate with methylated lysine. Examples of characterized domains are: tudor domain, MBT, WD40, ADD, zing finger CW domain, BAH, Ankyrin, PWWP, DCD, TTD and plant homeodomain (PHD) [60].

Lysine demethylases (KDM) are classified into two families: amines oxidase demethylases dependent on the flavin adenine dinucleotide (FAD) (example: KMD1A) and the family of the Jumonji domain proteins [60]. LSD1 acts on lysines 4 and 9 of mono- and dimethylated histone H3 [65]. LSD1 is overexpressed in several cancers. In acute lymphoblastic leukemia cells, LSD1 is overexpressed and mono- and dimethylation of lysine 4 of histone H3 is decreased. LSD1 silencing in these cells results in upregulation of H3K4me1 and H3K4me2 associated with induced apoptosis and inhibition of cell proliferation [66]. Members of the family containing the Jumonji domain interact with Fe (II) and α-ketoglutarate. Lysines 4, 9, 27 and 36 of histone H3 and lysine 20 of histone H4 are targeted by this family [67].

### 3.3. Main Active and Repressive Histone Marks

The combination of different PTMs deposited on histones defines regulatory functions such as enhancers, active promoters, repressed or chromatin in a bivalent state [68]. Bivalent domains define transcriptionally silent loci where both activating and repressive marks are “ready” to be activated during commitment to differentiation. The main histone marks activating or repressing transcription are listed in Table 2.

#### 3.3.1. Repressive Marks

Regarding methylation, the characteristic repressive marks correspond to the trimethylation of H3K9 and H3K27 [70]. H3K9me2 and H3K9me3 are associated with heterochromatin formation [71]. Readers, such as HP1, bind to H3K9me3 through their amino-terminal chromodomain and promote maintenance of the heterochromatin [72,73]. H3K9me3 marks both the constitutive heterochromatin where it is recognized by HP1 but also the facultative heterochromatin in the process of X chromosome inactivation [74]. The repressive H4R3me2 mark is catalyzed by PRMT5. This mark allows the recruitment of DNA methyltransferases (DNMT) [75]. H4K20me3 is localized in constitutive heterochromatin regions and silent genes [76].

An implication of these repressive histone modifications was studied in tumoral context. In this case, a disorganization of the three genomic compartments is found with a displacement of compartment B from the nuclear periphery to the interior of the nucleus. The preferential organization of the compartments is lost and heterogeneously distributed throughout the nucleus. Colon tumor cells show downregulation of genes in compartments B and I associated with block hypomethylation. In compartment I, downregulated genes are found enriched in H3K27me3 and those in compartment B by H3K9me3 and/or by methylation at the promoter. These mechanisms of block hypomethylation of B and I compartments are also found during successive passages and are characteristic of cells that have acquired excessive division [15].

#### 3.3.2. Active Marks

The H3K4 methylation mark is associated with transcriptionally active genes. Enhancers are marked with H3K4me1, active genes by H3K4me2 and active promoters and transcription initiation sites with H3K4me3. The presence of H3K4me1 increases the binding of the chromatin remodeling complex BAF at enhancers [77,78]. MLL, having a KMT activity with a SET domain, is a family of protein responsible for the methylation of H3K4 [79]. Acetylation of H3K27 is associated with active promoter and enhancers [80]. H3K9ac is also enriched at active gene promoters [81] (Figure 5).

Methylation of H3K79, localized in the globular domain of H3, is an activating mark and DOT1l catalyzes its mono-, di- or trimethylation and appears to be the only enzyme catalyzing this reaction for this residue [82,83]. Trimethylation of H3K79 is enriched in transcription start sites [84] and associated with transcriptional activity [85]. H3K36me3 is enriched at gene bodies [86].

H4R3me2a is associated with a decondensed chromatin state promoting histone acetylation. In the case of oxidative stress, OGG1 recruits the arginine methyltransferase PRMT1 and promotes the increase in the H4R3me2a mark at the promoter of c-Myc. The increase in H4R3me2a allows the recruitment of YY1 and promotes c-Myc gene transcription in response to oxidative stress [87].

## 4. Histone Modification Analysis Techniques

Different techniques can be used to characterize histone marks. Histone post-translational modifications can be studied using global quantification methods such as western blot, colorimetric or fluorometric techniques and mass spectrometry, based on the observation that a PTM of histones changes its molecular weight [88,89]. Analysis of writer and eraser activity is an indirect analysis of histone PTMs.

In addition, many techniques are available to study the distribution of post-translational modifications at a genome-wide scale and to define their target genomic loci. These newer high-throughput techniques are used to analyze the distribution of histone PTMs and associated chromatin structure in two main ways: chromatin accessibility by assay for transposase-accessible chromatin using sequencing (ATAC-seq) or targeted chromatin mapping, such as chromatin immunoprecipitation-sequencing (Chip-seq), CUT and RUN or CUT and TAG [90,91,92,93].

### 4.1. ATAC-Seq Method

The use of ATAC-seq does not require any prior knowledge of the epigenetic mechanism that regulates a given genomic region as it does not rely on the use of antibodies directed against a specific histone mark [91]. ATAC-seq is based on the use of the hyperactive prokaryote Tn5 transposase. After cell lysis, the transposase will digest DNA by cutting and inserting adapters (“cut-and-paste” mechanism) necessary for high-throughput sequencing within accessible chromatin regions (depleted of nucleosomes). The length of fragments obtained by Tn5 digestion is dependent on the spacing between nucleosomes [94]. Through this approach, “nucleosome-free regions” and footprints are defined. Footprints correspond to smaller fragments due to the binding of transcription factors on the DNA, causing a hindrance to transposase cleavage. ATAC-seq determines the positioning of nucleosomes, transcription factor binding sites and gene regulatory regions to be identified to the nearest nucleotide [95].

### 4.2. Targeted Chromatin Mapping Methods

#### 4.2.1. ChIP-Seq

Chromatin immunoprecipitation (ChIP), followed by high-throughput sequencing (seq), is now a commonly used method for genome-wide identification of specific post-translational modification-associated loci. ChIP-seq relies on the use of specific antibodies to determine the binding site of a target protein at a genome-wide scale. The antibodies can be directed against transcription factors to identify *cis*-regulatory sequences, but also against histone PTMs.

ChIP can be performed on native (N-ChIP) or cross-linked (X-ChIP) chromatin. X-ChIP consists in stabilizing DNA-protein interactions by bridging with chemical agents such as formaldehyde. The chromatin is then fragmented by ultrasound or enzymatic digestion. The accessibility of chromatin to the micrococcal nuclease varies according to the cell type, and different batches of enzymes may present variable activity, which may lead to biases. The size of the fragments obtained, following fragmentation, is important since it correlates with the resolution of ChIP [90].

#### 4.2.2. CUT and RUN

Cleavage Under Targets and Release Using Nuclease is an another high-throughput technique more recent than ChIP or ATAC-seq. This technique maps the interactions between DNA and proteins, but, unlike ChIP-seq, does not require pre-labelled fragmentation of the DNA and tends to better preserve DNA/protein interactions. Therefore, this method can be used to analyze the binding of transcription factors to their *cis*-regulatory regions but also to analyze PTMs of histones [92]. No cross-linking of DNA/protein interactions is required. CUT and RUN uses specific antibodies to which the protein A micrococcal nuclease (pA/Mnase) is attached. This will be activated by the addition of Ca^2+^ and will allow the cleavage on both sides of the protein binding site. The DNA fragments will be submitted to sequencing by sequencing adapter ligation [96].

#### 4.2.3. CUT and TAG

This technique also uses antibodies specifically directed against transcription factors or PMTs of histones. The Tn5 transposase, also used in ATAC-seq, is coupled to protein A. This fusion protein will guide the Tn5 transposase to the binding site of the antibody directed against the protein of interest. The transposase is coupled with adapters for subsequent sequencing [93].

The coupling of different techniques allows the visualization of different activating or repressive histone marks correlated with the chromatin opening state [97,98].

## 5. Histones and Neuronal Differentiation

Post-translational modifications of histones in the modulation of chromatin compaction and access to the transcriptional machinery are involved at the early stages of development, and particularly during neuronal differentiation.

### 5.1. Overview of Neuronal Differentiation

During embryogenesis, the neuroepithelium gradually expands to form the neural tube. The cells lining the neural tube are neural stem cells (NSC). These NSCs are multipotent, i.e., able of self-renew and differentiate. They allow the development of the central nervous system, including the spinal cord and brain by differentiating to neuronal, astrocytic and oligodendrocytic lineages [99]. In embryonic mammalian brains, neuroepithelial cells from the cortex will first divide symmetrically and later differentiate into radial glial cells (RGCs) through asymmetrical divisions allowing their self-renewal [100]. RCGs are located in the ventricular zone of the cortex, characterized by a loss of tight junctions and express astrocytic markers. RGCs are considered the neural progenitor cells (NPCs) of the ventricular zone [99] that will give rise to other cell subtypes, notably the intermediate progenitors. These intermediate progenitors, located in the subventricular zone, will in turn divide to give rise to mature cortical neurons [101]. The differentiation process of the central nervous system is a temporal and spatial regulation mediated by molecular cues and interactions with the microenvironment. Glial cell differentiation occurs at the end of neurogenesis. The generation of glial cells promotes the proper functioning of established neural networks (Figure 6) [102].

Neurogenesis is not limited to the embryonic stage. Neurogenesis also occurs in the subventricular zone (SVZ) of the lateral ventricles and the subgranular zone (SGZ) of the dentate gyrus of the hippocampus in the adult mammalian brain [5,103,104]. These neurogenesis sites contain adult stem cells, coming from a population of radial glial cells that did not differentiate during embryogenesis [105]. The tri-potent potential of these adult radial glial cells remains controversial. Inactivation of neurofibromin 1 (*Nf*1) in radial glial cells from the dentate gyrus of adult mice restores this tripotent potential by generating the three neural lineages [106]. Adult neurogenesis can be increased in the SVZ by expression of Cdk4/cyclinD1 in mice increasing generated neuron integrating the olfactory bulb. This increased neurogenesis improves odor discrimination [107].

### 5.2. Models for Studying Histone Modifications during Neuronal Differentiation

Animal models (e.g., zebrafish, mouse) as well as human induced pluripotent stem cells (hIPSC) have made it possible to overcome the limited inaccessibility of human neurons to study epigenetic mechanisms during neuronal differentiation. Several protocols have been developed to obtain NPCs from hIPSC with neuronal rosettes obtained in cultures resembling the neural tube. These NSCs can be differentiated into different types of neurons (glutamatergic, dopaminergic, GABAergic, etc.) using appropriate protocols [108,109,110]. In addition, it is now possible to generate a three-dimensional structure such as organoids or assembloids allowing a better view of cellular processes and cell–cell interactions. These approaches can also be considered to improve the understanding of the pathophysiology of chromatinopathies as reviewed in [111].

### 5.3. Histone Changes

The mechanisms of NSCs differentiation are spatio-temporally regulated by changes in the accessibility of transcription factors to the promoter of genes that regulate the differentiation to a specific lineage. Changes in chromatin will gradually limit access to multipotency genes, with downregulation of pluripotency genes (*OCT*4, *NANOG*) and promote access to the specific differentiation program associated with an increase in NPC markers (*PAX*6, *SOX*1 et *OTX*2) [112].

#### 5.3.1. Role of Histone Acetylation and Deacetylation

HDAC are essential for the maintenance of self-renewal and proliferation of the neural stem cell by maintaining, through their catalytic activity, transcriptional suppression on target genes. Several studies have been conducted to elucidate the role of HDACs in neuronal differentiation. Transient and incomplete HDAC inhibitors such as 2-propylpentanoic acid (called valproic acid or VPA), trichostatin A (TSA), suberoylanilide hydroxamic acid (SAHA) and sodium butyrate (NaB) have been used for this purpose. HDAC inhibition is associated with histone hyperacetylation and excessive inhibitor concentrations, leading to cell death [113].

The use of VPA induces neuronal differentiation by affecting histone acetylation. In rat embryonic hippocampus neural progenitor cells, VPA reduces cell proliferation and initiates neuronal differentiation. This is mediated by an increase in the expression of *Ngn*1, *Math*1 and *p*15 through an increase in H4 acetylation at their promoter. Similar results were observed in vivo in mice where VPA-treated animals showed an increase in H4 acetylation at the *Ngn*1 promoter inducing its expression [7]. In NSCs from rat embryo hippocampi collected at E16 and treated with VPA, induction of neuronal differentiation is associated with an increase in the H3K4me3 and H3K9ac active marks at the *Ngn*1 promoter together with a decrease in the repressive H3K27me3 and H3K9me3. These modifications are independent of the AKT/mTOR pathway activated after VPA treatment [114].

VPA also promotes neuronal differentiation in H9 human embryonic stem cells (HUES). Addition of VPA to the differentiation medium increases the expression of neuronal markers (β3-tubulin, MAP2, NEUN) and those specific to the GABAergic and dopaminergic subtypes. The markers of mature neuron, NEUROD1, and NeuN are expressed 30-times higher after VPA addition. This increase is also observed for oligodendrocyte markers (MBP) but not for astrocyte markers (S100β). Differentiation toward the astrocytic lineage is decreased under these conditions, which is usually observed during differentiation to the neuronal lineage. Length and connections established by neurites are also significantly increased by VPA addition [115]. Besides, the use of retinoic acid (NaB) increases the acetylation of H3K9 at the *PAX*6 promoter, which promotes the induction of neural progenitor cells [116].

H3K9ac is also involved in the kinetics of neural differentiation. Qiao et al., showed that during the first four days of neural differentiation of HUES-9, H3K9 acetylation decreases and then gradually increases between days four to eight. The kinetics of H3K9 acetylation at specific promoters allows the silencing of pluripotency genes and the expression of early neural genes. This is supported by experiments on p300 and the use of TSA. Indeed, shRNA depletion of p300, involved in H3K9 acetylation, inhibits the expression of early neural genes (*PAX*6, *SOX*1, *ZIC*2 and *ZNF*521). In addition, the use of TSA inhibits the expression of early neural genes, *PAX*6 and *SOX*1, between days zero to four but increases their expression between days four to eight. This is evidenced by an increase in the proportion of NPCs. TSA inhibits HDAC1/5 and 8, which allows the increase in H3K9ac and the differentiation to the neural lineage. The expression of early neural genes such as *PAX*3/6, *SOX*1, *POU*3*F*2 is improved by HDAC1/5 and 8 knockdown, while pluripotency genes are downregulated. These three enzymes allow an appropriate neural differentiation by preventing an early differentiation [112].

#### 5.3.2. Role of Methylation and Demethylation

During in vitro neuronal differentiation from murine embryonic stem cells to neural progenitor cells, significant opposite changes are observed for H3K27ac and H3K79me2 through genome-wide acting mechanisms. H3K27ac progressively decreases by a factor of two between the two stages, while H3K79me2 is increased four-fold, genome-wide. A gain of H3K79me2 during development is observed for a group of genes with a clear axonogenic signature and critical for neuronal development. For these genes, H3K79me2 correlates with an increase in gene expression. A decrease in H3K27ac indicates a progressive chromatin condensation during neuronal differentiation in this model. These global opposing changes of histone PTMs might be the consequence of two mechanisms: the accumulation of histone marks at specific loci with a global shift of the number and magnitude of these marks depending on the conditions associated with a genome-wide global gain/loss of the signal due to a balance of accumulation/deletion of histone PTMs [117].

Similar results were also observed when mouse ESCs were transduced into NPCs. The mESCs are in a more open chromatin conformation than differentiated cells with a higher proportion of active marks such as H3K4me3, H3K9ac and H3K14ac compared to NPCs [118]. The differentiation of mESCs is associated with an increase in H3K9me3 and a decrease in acetylation of H3 and H4 [119]. Neuronal differentiation in vitro is thus accompanied by a deacetylation associated with a more compact chromatin compared to mESC [117].

On the other hand, genes associated with differentiation and development are found in a bivalent state with trimethylated marks of H3K27 and H3K4 at the mESC stage. These bivalent domains are reduced in differentiated cells and developmental genes acquire either repressive or activating marks. For genes induced during neural differentiation (e.g., *Nkx*2.2, *Sox*21 and *Zfpm*2), in a bivalent conformation in mESCs, a resolution of the bivalent state by methylation of the transcription start site H3K4 is observed [120]. In NSCs, genes that are specific to the neuronal lineage are kept in a silent state and expressed later during differentiation after the loss of the repressive H3K27me3 mark [121].

DOT1L is currently considered the only enzyme catalyzing H3K79me methylation during neuronal differentiation [82,83]. Its inhibition in mECSs and neural progenitors decreases the H3K79me2 marker and promotes neuronal differentiation in mESC-derived NPCs. This treatment leads to a decrease in neural stem cell markers in NPCs associated with an increase in markers of fully differentiated neurons. DOT1L inhibition alters the distribution of H3K27ac in ESCs and NPCs. Gene promoters are also affected. Genes associated with promoters in a repressed state (H3K27me3) recognized by Polycomb complexes or promoters in a bivalent state (H3K4me3 and H3K27me3) are both upregulated. Conversely, active genes (H3K4me3 and H3K27ac enriched-promoters) are downregulated following DOT1L inhibition. DOT1L inhibition also reduces SOX2 binding to its target enhancers in NPCs. *SOX*2 expression is maintained in both embryonic and adult neural stem cells until differentiation [117,122]. The decreased binding of SOX2 to its target enhancers might explain the neuronal differentiation of NPCs following DOT1L inhibition. DOT1L would therefore be involved in the maintenance of NPCs by preventing their differentiation via SOX2 [13,117].

Another KMT, MLL1 plays an important role in neuronal differentiation by methylating H3K4 [123] and by being involved in the demethylation of H3K27 by recruiting demethylases [124]. This role was illustrated by studying the neuronal differentiation of the SVZ olfactory bulb from SVZ (subventricular zone) monolayer NSC cultures derived from an M*ll*1-deficient mouse model. This model shows normal astrocytic and oligodendrocytic cell differentiation but impaired neuronal differentiation with a significant decrease in the transcription factor DLX2 involved in terminal differentiation of interneurons. MLL1 binds to the promoter of the *Dlx*2 gene. In wild-type mice NSCs, activating H3K4me3 marks are enriched at the *Dlx*2 promoter. In *Mll*1-deficient mice in which Dlx2 expression is impaired, no decrease in H3K4me3 is reported but a significant increase in H3K27me3 is observed. In these mice, the *Dlx*2 locus becomes bivalent, and gene transcription is repressed. *Mll*1 thus allows the removal of bivalent marks allowing the induction of neuronal differentiation by modulation of gene expression [125]. Studies on whole zebrafish embryos during the development of the nervous system have shown that morpholino oligonucleotide-mediated MLL1 protein depletion is associated with a decrease in the number of SOX2 positive NPC due to a defect in their proliferation. No proliferation defect was observed in embryonic neurons. However, this decrease in MLL1 is associated with early neuronal differentiation and also negatively affects gliogenesis [126]. This leads to a decrease in the expression of the neural progenitor marker *neurogenin*1 (*Ngn*1) and upregulation of postmitotic neuron-specific markers.

The EHMT2 methyltransferase is involved in the demethylation of mono- and dimethylated H3K9 [127] and also in the methylation of H3K27 residues [128]. This enzyme has two isoforms that differ by the incorporation or not of exon 10 [129]. EHMT2 activity is essential for neuronal differentiation. Indeed, EHMT2 knockdown in the mouse-neural-crest-derived cell line (N2a) abolishes neurite growth and inhibition of its methyltransferase activity preventing the acquisition of a fully differentiated phenotype. The essential role of the catalytic activity of EHMT2 is underlined by the rescue of the phenotype by transfecting wild-type EHMT2, which is not possible with the mutant without any catalytic activity. During neuronal differentiation of N2a, expression of the EHMT2 isoform with exon 10 and H3K9 dimethylation are increased. Consistently, knockdown using an siRNA directed against exon 10 shows an inhibition of neuronal differentiation indicating that this isoform is necessary for neuronal differentiation [130].

The use of a morpholino antisense oligonucleotide against the *Sox*19*b* transcription factor during embryonic development in zebrafish, leads to a decrease in H3K27me3 at the *Ngn*1 and *ascl*1*a* promoters, without affecting the histone acetylation state. This decrease is due to a decreased expression of EZH2, that catalyzes H3K27 trimethylation. Premature entry of neural tube NSCs into neuronal differentiation is then induced by an increase in the level of *Ngn*1 and *ascl*1*a* transcription, showing a role for the repressive H3K27me3 mark during neuronal differentiation. Thus, the elimination of this repressive marker controls the balance of self-renewal/differentiation of NSCs in zebrafish. The importance of a proper functioning of this balance is further highlighted by the neural tube abnormalities associated with a decrease in the surface area of the diencephalon and telencephalon in SOX19B-deficient embryos [131].

Lentiviral-induced overexpression of the neurogenic factor NeuroD1 is able to reprogram mouse microglia and oligodendrocytes into mature neurons both in vitro and in vivo. These neurons are β3 tubulin-positive and respond to N-methyl-D aspartate (NMDA) stimulation. NeuroD1-mediated conversion proceeds directly from microglia to mature neurons without intermediates such as neural stem cells or *Nestin*-positive precursors. The generated neurons are predominantly glutamatergic and form excitatory synapses. NeuroD1 initiates this process of neuronal differentiation by binding to bivalent chromatin regions (enriched in H3K4me3 and H3K27me3) present in microglia at genes involved in neuronal development and differentiation such as *Bhlhe*22, *Brn*2 or *Pou*3*f*3. Genes upregulated after NeuroD1 binding show a decrease in H3K27me3 and an increase in H3K4me3. Furthermore, microglial identity is suppressed by modification of histone marks at the promoters of microglia-specific genes. A decrease in H3K4me3 and an increase in H4K27me3 on its promoter regions is observed [6].

#### 5.3.3. Modifications of 3D Architecture of Chromatin during Neuronal Differentiation

As previously discussed, 3D compartmentalization that plays a role in the maintenance and regulation of the epigenome and by corollary, also influences neuronal differentiation processes. Comparison of Drosophila embryonic and neuronal cells indicates that three-dimensional organization is modulated during neuronal differentiation with dynamic changes in enhancer/promoter interactions [132]. A number of TADs that are specific and common to both lineages have been identified. Neuron-specific TADs show enrichment in enhancers driven by neuron-specific transcription factors [132]. The regulation of A/B compartmentalization, TADs and enhancer/promoter loops in neural development is reviewed by Kishi et al., Arzate-Bejía et al. and Ghosh et al. [133,134,135]. In particular, interactions within compartments A and B are modulated during in vitro neuronal differentiation. A decrease in interactions within compartment A is observed during the first stage from mESCs into NPCs. Then, interactions within compartment B increase during the transition from NPCs to cortical neurons [136].

For a more global approach for understanding neuronal differentiation, it is now possible to reconstruct the 3D organization of the genome using predictive methods. An improved population-based modeling approach and a probabilistic framework to model a population of 3D structures of entire diploid genomes was tested on human lymphoblastoid cells. The predicted 3D structures correctly identified many of the features of the lymphoblastoid genome obtained from imaging experiments, including interchromosomal distances between gene loci, their interactions and preferred nuclear locations of chromosomes [137]. Histone PTMs, together with sequence information of CTCF binding sites, are able to predict chromatin structure at a resolution from 5kb. This elaborate model combining both bioinformatics analysis with polymer modeling has allowed the characterization of TADs, compartments and chromatin loops [138]. Other methods for modeling the 3D structure of the genome are available, notably the data-driven polymer model, which allows the summarization of the preferential chromosomal positions, the A/B compartmentalization, to distinguish the inactive X chromosome by a more important compaction and a more inward localization than its active homologue [139].

## 6. Pathological Involvement of the Epigenetic Machinery

A number of neurodevelopmental disorders and cancers result from mutations in components of the epigenetic machinery [4,140,141,142]. Alterations in the functions of writers, readers and erasers or chromatin remodelers have been reported as associated with Mendelian disorders. To date, 86 diseases resulting from mutations in 74 genes encoding chromatin-enzymes have been reported [3,143,144]. Such disorders are called “chromatinopathies” and are listed in Table 3.

### Chromatinopathies and Neurodevelopment

Among rare diseases, chromatinopathies account for 5% to 10% of Mendelian disorders [145]. More specifically, 64 out of the 86 listed diseases (74.4%) are associated with neurodevelopmental disorders, including attention-deficit/hyperactivity disorder, communication disorders, autism spectrum disorder, neurodevelopmental motor disorders and intellectual disability [146]. This illustrates the crucial importance of epigenetic control in the homeostasis and plasticity of neuronal development. Therefore, a better understanding of these mechanisms represents a real challenge for the improvement of diagnostic yield and the development of therapeutic tools for a number of these chromatinopathies.

Numerous models have been developed for the study of chromatinopathies and in particular those leading to neurodevelopmental disorders [147]. Mouse models of Kabuki syndrome 1, mutated in *Kmt*2*d*, present an alteration of neurogenesis with a decrease in DCX+ cells, a marker of immature neurons. These mice exhibit memory defects and visuospatial learning deficits associated with H3K4 hypomethylation in the hippocampus. These defects could be reversed by the use of TAK-418, a KDM1A inhibitor [148]. Regarding Rett syndrome (RTT; OMIM #312750,), the *Mecp*2-mutated mouse model revealed that an improvement of the phenotype is observed after restoration of systemic administration of MECP2 [149]. Furthermore, these models show that intensive pre-symptomatic training could delay the onset of symptoms, by improving the morphology of the hippocampal granules and cortical neurons as well as electrophysiological defects [150]. In Rubinstein Taybi syndrome heterozygous *Cpb*^+/−^ mice, the defect in *CREBBP* HAT activity results in long-term memory defects associated with H2B hypoacetylation in the hippocampus [151]. Similar results have been observed in lymphoblastoid cells of Rubinstein Taybi syndrome patients with hypoacetylation of histones H2A and H2B [152].

The development of hiPSC-based models represents an interesting alternative to animal models by mimicking steps of human neuronal development [153,154,155]. The generation of hiPSC-derived neurons from Rubinstein Taybi syndrome patients (iNeuron) revealed morphological abnormalities with a reduced size but increased number of neurites and hypoexcitability [156]. In this model, RNA-seq transcriptomic analyses reveal disrupted transcriptional regulation at several levels in RSTS patients compared to controls during neuronal differentiation with alteration of genes involved in neuronal migration and axonal and dendritic functions. Finally, a decreased modulation of the overall number of neural differentiation markers may explain the defects in neuronal differentiation in these patients [157]. Similarly, a quantitative proteomic analysis of Rett iPSC-derived neuronal progenitors revealed alterations for proteins involved in dendrite morphology and synaptogenesis pathways during early neuronal differentiation [158].

The complexity of the human brain and neuronal development cannot be limited to single layer models. To mediate these issues, 3D models have been developed. Brain organoids reproduce part of the development of the nervous system and model the sequential generation of neurons and glial cells [159]. Interestingly, during differentiation of cerebral organoids, the epigenome is similar to that of the human fetal brain.

Indeed, it was shown that methylated cytosines outside of CG contexts, called mCH, were similar between brain organoids at Day 40 and Day 60 of differentiation and mid-fetal cortex. The accumulation of mCH in cerebral organoids and the fetal cortex is associated with loci that overlap with putative super-enhancers. Corresponding regions are actively expressed during early fetal stages and then become repressed during the neonatal/postnatal transition. In addition, comparative transcriptomic analyses showed a strong correlation between brain organoid differentiation and the human mid-fetal neocortex. The genetic pathways are similar to those of the fetal cortex, including NPC proliferation and self-renewal, migration, adhesion, delamination and differentiation suggesting that in vitro organoids tend to recapitulate embryonic development [160,161]. Organoids from different brain regions have been used to model Rett syndrome. Premature development of the cortical plate, together with an increased number of neurons generated, associated with an early depletion of the proliferating progenitor pool, are observed in dorsal forebrain organoids [162].

The large phenotypic overlap between different chromatinopathies (e.g., RSTS, KS, Wiedmann-Steiner (WDSTS; OMIM #605130), Cornelia de Lange (CdLS, OMIM #122470, #300590, #610759, #614701, #300882, #608749)) can be explained by epigenetic modulations occurring within interconnected networks [3,163,164]. This results in diagnostic challenges both for clinical orientation and interpretation of variants of unknown significance (VUS) in genes encoding chromatin-modifying enzymes [165]. The emergence of specific DNA methylation episignatures provides new diagnostic insights when new generation genome sequencing techniques have not allowed a certainty diagnosis [144,166]. To date, 57 episignatures have been described for 65 genetic syndromes. Neurodevelopmental disorders are mostly represented with 38 specific signatures for 43 syndromes [167]. These episignatures are also extended to imprinting disorders such as Prader Wili syndrome [145,168]. Regarding chromatinopathies, episignatures focusing on PTMs of histones could be more specifically developed. Then, the challenge will be to integrate these new biomarkers in the context of diagnostic testing.

Based on a better understanding of the pathophysiological mechanisms leading to these pathologies, new therapeutic perspectives have emerged. HDAC inhibitors have shown promising results in Rubinstein Taybi syndrome, being able to reverse the memory defects in mouse models, and to improve morphological and electrophysiological defects of iNeurons [151,169]. Furthermore, in Kabuki syndrome, defects in H3K4 trimethylation, neurogenesis and memory were normalized by HDAC inhibitor treatment (AR-42) [170]. As well as pharmacological therapeutics, gene therapies have been considered for nervous system pathologies. Viral-mediated gene therapy vectors such as AAV have the advantage that some specific serotypes can cross the blood-brain barrier after systemic injection [171]. The question of the amount of genetic material provided is to be considered for this kind of therapy. For example, in Rett syndrome, the duplication of MECP2 leads to neurological disorders such as motor dysfunctions and intellectual disabilities, so the right amount of protein must be determined [172]. Therapy using AAV2 ITR-flanked genomes packaged into AAV9 serotype capsids to transfer wild-type MECP2 improves survival of Mecp2-null mice and delays pathology progression [173]. Furthermore, combined approaches have been developed for Rett syndrome, using both antisense oligonucleotides directed against XIST RNA and a small molecule inhibitor of DNA methyltransferase, to selectively reactivate the inactivated X chromosome and report the missing protein amount. MECP2 is found reactivated in this context up to 30,000 of Xi levels [174]. This approach could be considered for other X-linked diseases due to protein quantity defects. A possible future therapeutic approach is to use the CRISPR tool to directly correct the mutation at the gene level. A demonstration of the potential of this method has been observed in the treatment of Duchenne muscular dystrophy (DMD, OMIM #310200), where the use of CRISPR/Cas9 in patient’s myoblasts restored the expression of dystrophin [175]. This technology has also been used to model neurodevelopmental disorders by introducing a specific mutation into a brain organoid model [176]. An alternative version using the dead cas9 nuclease has been used in fragile X syndrome (OMIM #300624, FXS). The dead cas9 fused to Tet1 together with a target sequence allows demethylation of CGG repeats in the *FMR*1 locus, which is hypermethylated in this disease. This leads to a silencing of the *FMR*1 gene in IPSCs maintained in derived neurons [177]. This approach could be extended to other neurodevelopmental disorders characterized by hypermethylation at a specific locus. Because of the reversible nature of epigenetic modifications, all components of the epigenetic machinery constitute a potential therapeutic target, however, keeping in mind the possible genome-wide effect of non-targeted epi-therapies.

## 7. Conclusions and Perspectives

Epigenetic mechanisms have been shown to be essential for modulating gene expression and enabling the acquisition of cellular identity. Here, we focused on the importance of epigenetic modulations in neuronal differentiation, but also on the pathological consequences associated with deficits in the epigenetic machinery. Neurodevelopmental disorders represent about 75% of chromatinopathies [3,166]. A better understanding of the epigenetic mechanisms involved in neuronal differentiation could lead to a reduction in the diagnostic deadlock in these pathologies. The emergence of DNA methylation episignatures has already allowed new diagnostic advances, more than half of which concern neurodevelopmental disorders [166].

A multi-omics integration of these episignatures coupled with data from transcriptomic and chromatin dynamics high-throughput techniques would allow to refine the pathophysiology and the biological pathways deregulated in these chromatinopathies. The joint development of brain organoid models [178] allows new therapeutic perspectives.

## Figures and Tables

**Figure 1 genes-13-00639-f001:**
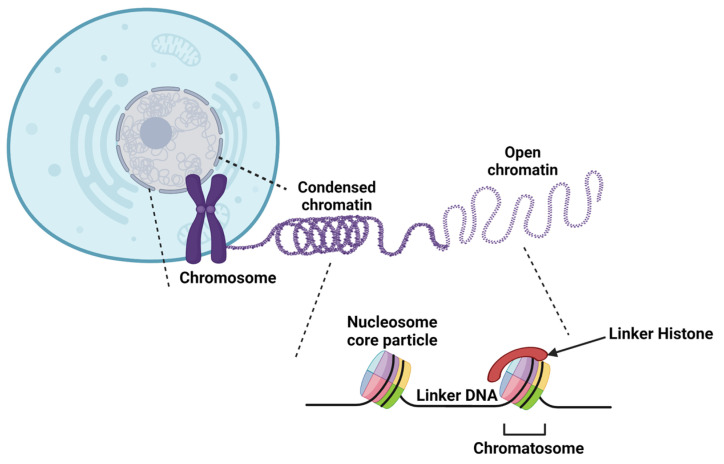
Overview of the different levels of DNA compaction. The first higher-order structure of the chromatin is the nucleosome, which is composed of 145–147 bp of DNA wrapped around a H3/H4 tetramer and two H2A/H2B dimers. The histone H1 linker binds DNA fragments that link two nucleosomes to form the chromatosome. The addition of H1 promotes internucleosomal interactions and the formation of the 30 nm chromatin fiber. The highest level of DNA compaction is the metaphasic chromosome, observable during cell division.

**Figure 2 genes-13-00639-f002:**
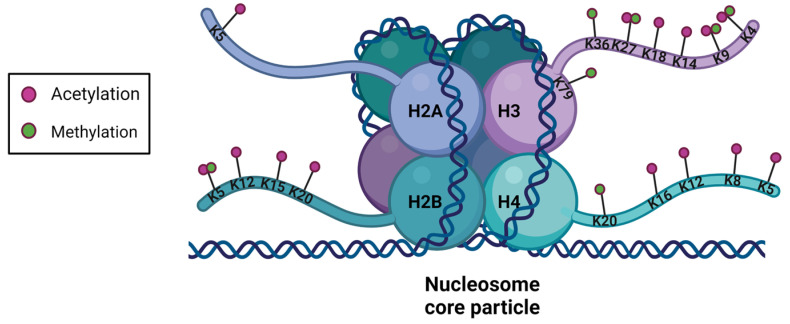
Structure of a nucleosome and main sites of methylation and acetylation in histones. Post-translational modifications of the histones are mostly performed on the amino-terminal tails of the histones accessible to the epigenetic writer and eraser. Acetylated residues are in pink and methylated ones are in green.

**Figure 3 genes-13-00639-f003:**
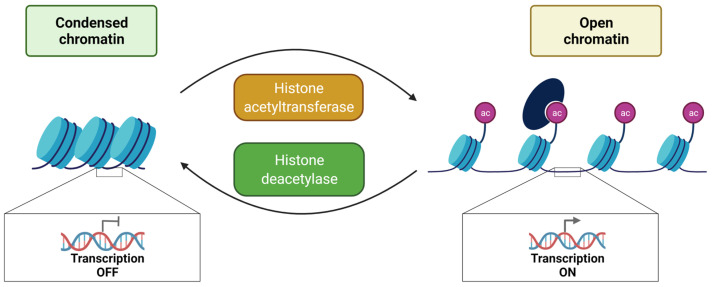
Overview of histone acetylation and deacetylation and their impact on chromatin structure. A histone acetylation reader protein is shown in dark blue. Bromodomain-containing proteins can bind to acetylated histones.

**Figure 4 genes-13-00639-f004:**
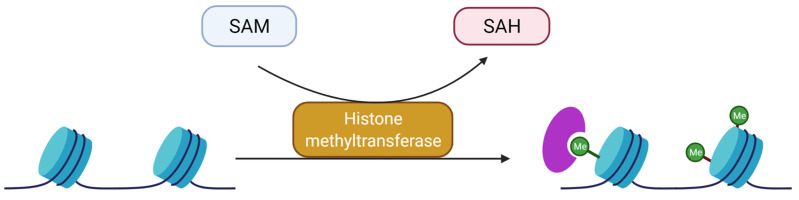
Overview of the mechanism of histone methylation by a methyltransferase. S-adenosyl-L-methionine (SAM) is the methyltransferase cofactor. Following methylation, S-adenosylhomocysteine (SAH) is released and the methylated moiety is attached to Nε-Lysine. A methylation reader is represented in purple and corresponds, for example, to a chromodomain or tudor domain protein.

**Figure 5 genes-13-00639-f005:**
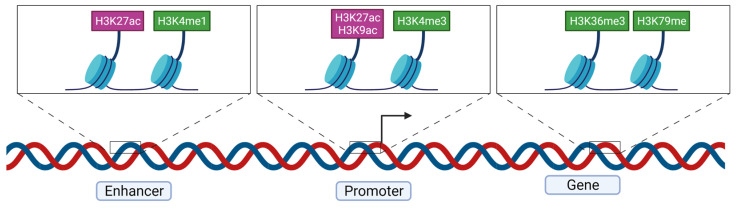
Major histone marks found at active enhancers, promoters and gene bodies. The arrow represents the transcription start site. H3K27ac is found both at active enhancers and promoters. H3K4me1 is enriched in enhancer regions whereas H3K4me3 is found mostly at promoters. Active gene regions are enriched in H3K36me3, H3K79me2 and H3K79me3.

**Figure 6 genes-13-00639-f006:**
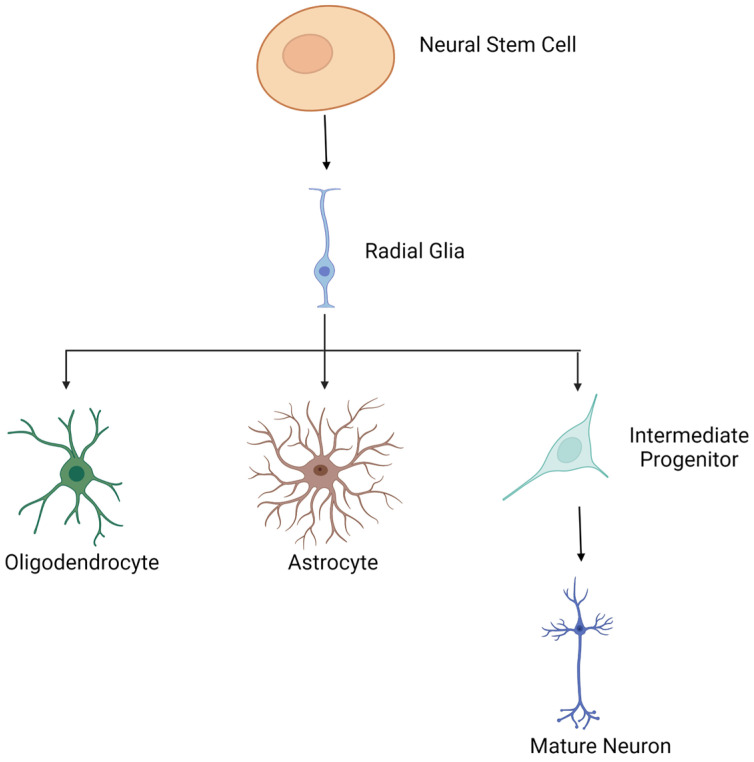
Neural stem cells give rise to the neuronal and glial lineage by passing through the neural progenitor stage. Neurogenesis occurs in the embryonic brain but also in the adult brain.

**Table 1 genes-13-00639-t001:** Histone acetylation and deacetylation are performed by KATs and HDACs, respectively. Two types of KATs are defined: type A and B. Type A KATs have a nuclear localization and modify chromatin-associated histones, whereas type B KATs, mainly localized at the cytoplasmic level, modify newly synthesized histones not incorporated in the nucleosome. KATs are classified into six families and HDACs into four.

**Histone** **Acetyltransferase**	**TYPE B**	**TYPE A**				
	**Cytoplasmic KAT Family**	**GNAT Family**	**P300/CBP Family**	**MYST Family**	**Transcription Factor Related Family**	**Nuclear Receptor** **Coactivator Family**
	KAT1 (HAT1)	KAT2A (hGCN5)	KAT3A (CBP)	KAT5 (TIP60/PLIP)	KAT4 (TAF1)	KAT13A (SRC1)
		KAT2B (PCAF)	KAT3B (p300)	KAT6A (MOZ/MYST3)	KAT12 (TFIIIC90)	KAT13B (ACTR)
		KAT9 (ELP3)		KAT6B (MORF/MYST4)		KAT13C (P160)
				KAT7 (HBO1/MYST2)		KAT13D (CLOCK)
				KAT8 (HMOF/MYST1)		
**Histone Deacetylase**	**Class I**	**Class II**		**Class III**	**Class IV**	
	HDAC1	HDAC4		SIRT1	HDAC11	
	HDAC2	HDAC5		SIRT2		
	HDAC3	HDAC6		SIRT3		
	HDAC8	HDAC7		SIRT4		
		HDAC9		SIRT5		
		HDAC10		SIRT6		
				SIRT7		

**Table 2 genes-13-00639-t002:** Main post-translational modifications of histones and targeted residues. Effects on gene expression are reported. Adapted from Lachat et al. (2018) [69].

Modified Histone Residue	Type of Post-Translational Modification	Target Amino Acid	Effect on Gene Expression
H2AS1	Phosphorylation	Serine, threonine, tyrosine	Modulation of DNA compaction and interaction with other histone post-translational modifications
H2AS139			
H2BS14			
H3T3			
H3T6			
H3S10			
H3T11			
H3S28			
H4S1			
H2AK119	Ubiquitination	Lysine	Regulation of transcription initiation and elongation
H2BK120			
H2AK5	Acetylation	Lysine	Decrease in histone/DNA interaction, chromatin is structurally loose, less compact and transcription is activated
H2BK5			
H2BK12			
H2BK15			
H2BK20			
H3K4			
H3K9			
H3K14			
H3K18			
H3K27			
H4K5			
H4K8			
H4K12			
H4K16			
H2BK5me1	Methylation	Lysine and arginine	Transcription is activated
H3K4me1			
H3K4me2			
H3K4me3			
H3K9me1			
H3K27me1			
H3K36me3			
H3K79me1			
H4K20me1			
H3K9me2	Methylation	Lysine	Transcription is inhibited
H3K9me3			
H3K27me3			
H3K79me3			
H3K18	SUMOylation	Lysine	Competition with other lysine modifications. Decrease and stop of transcription
H4K12			

**Table 3 genes-13-00639-t003:** Mendelian disorders of the epigenetic machinery. Writer, eraser and remodeler enzymes that also carry a reader domain are marked by an asterisk. Adapted from [3,143,144].

Gene Function	Gene	Disease
**Writer** Histone Acetyltransferase	*CREBBP* *	Rubinstein Taybi syndrome 1
	*EP*300 *	Rubinstein Taybi syndrome 2
	*KANSL*1	Koolen-De Vries syndrome
	*KAT*6*A* *	Mental retardation autosomal dominant (MRAD) 32
	*KAT*6*B* *	Say-Barber-Biessecker-Young-Simpson syndrome/Genitopatellar syndrome
**Writer** Histone Methyltransferase	*ASH*1*L* *	MRAD 52
	*EHMT*1 *	Kleefstra syndrome 1
	*EZH*2	Weaver syndrome
	*KMT*2*A* *	Wiedemann-Steiner syndrome
	*KMT*2*B* *	Childhood-onset dystonia 28
	*KMT*2*C* *	Kleefstra syndrome 2
	*KMT*2*D* *	Kabuki syndrome type 1
	*KMT*2*E* *	KMT2E deficiency
	*KMT*5*B*	MRAD 51
	*NSD*1 *	Sotos syndrome 1
	*NSD*2 *	Wolf-Hirschhorn syndrome
	*PRDM*5	Brittle cornea syndrome
	*PRDM*12	Hereditary sensory and autonomic neuropathy 8
	*PRDM*16	Dilated cardiomyopathy
	*SETD*5	MRAD 23
	*SETD*2	Luscan-Lumish syndrome
	*SETD*1*B*	SETD1B-related syndrome
**Writer** DNA methyltransferase	*DNMT*1 ***	AD cerebellar ataxia deafness and narcolepsy/Hereditary sensory neuropathy 1E
	*DNMT*3*A* *	Tatton-Brown-Rahman syndrome/Microcephalic dwarfism
	*DNMT*3*B* *	Immunodeficiency-centromeric instability-facial anomalies syndrome
**Eraser** Histone deacetylase	*HDAC*4	Brachydactyly–mental retardation syndrome
	*HDAC*6	Chondrodysplasia with platyspondyly/distinctive brachydactyly/hydrocephaly and microphthalmia
	*HDAC*8	Cornelia de Lange syndrome type 5
**Eraser** Histone demethylase	*HR*	Alopecia universalis/Atrichia with papular lesions/Hypotrichosis type 4
	*KDM*1*A*	Cleft palate, psychomotor retardation, and distinctive facial features
	*KDM*5*B* *	Mental retardation autosomal recessive 65
	*KDM*5*C* *	Claes–Jensen X-linked mental retardation
	*KDM*6*A*	Kabuki syndrome type 2
	*KDM*6*B*	KDM6B deficiency
	*PHF*8 ***	Siderius X-linked mental retardation
**Remodeler**	*ARID*1*A*	Mental retardation autosomal dominant 14 (Coffin–Siris syndrome)
	*ARID*1*B*	Mental retardation autosomal dominant 12 (Coffin–Siris syndrome)
	*ATRX **	α-thalassemia/mental retardation X-linked (ATRX) syndrome
	*CHD*1 ***	Pilarowski–Bjornsson syndrome
	*CHD*2 ***	Epileptic encephalopathy, childhood onset
	*CHD*3 ***	Snijders Blok–Campeau syndrome
	*CHD*4 ***	Sifrim–Hitz–Weiss syndrome
	*CHD*7 ***	Charge syndrome/Hypogonadotropic hypogonadism
	*CHD*8 ***	Autism susceptibility 8/overgrowth and ID
	*SMARCA*2 ***	Nicolaides–Baraitser syndrome
	*SMARCA*4 ***	Coffin-Siris syndrome 4/Rhabdoid tumor predisposition syndrome
	*SRCAP*	Floating–Harbor syndrome
**Reader**	*AIRE*	Autoimmune polyendocrinopathy syndrome 1
	*ALG*13	Eary infantile epileptic encephalopathy 36
	*ASXL*1	Bohring–Opitz syndrome
	*ASXL*2	Shashi–Pena syndrome
	*ASXL*3	Bainbridge–Ropers syndrome
	*BPTF*	Neurodevelopmental disorder with dysmorphic facies and distal limb anomalies
	*BRPF*1	Intellectual developmental disorder with dysmorphic facies and ptosis
	*BRWD*3	X-linked mental retardation 93
	*CBX*2	Sex reversal
	*DPF*2	Coffin Siris 7
	*EED*	Cohen–Gibson syndrome
	*LBR*	Pelger–Huet anomaly (PHA)/PHA with muskuloskeletal findings Greenberg skeletal dysplasia
	*MBD*5	MRAD 1
	*MECP*2	Rett syndrome and related disorders
	*MORC*2	Charcot–Marie–Tooth disease 2Z
	*MSH*6	Hereditary nonpolyposis colorectal cancer 5/Mismatch repair cancer syndrome
	*ORC*1	Meier–Gorlin syndrome 1
	*PHF*6	Borjeson–Forssman–Lehmann syndrome
	*PHIP*	Developmental delay, ID, obesity and dysmorphic features
	*RAG*2	Omenn syndrome and severe combined immunodeficiency
	*RAI*1	Smith–Magenis syndrome
	*RERE*	Neurodevelopmental disorder with or without other anomalies
	*SMN*1	Spinal muscular atrophy
	*SP*110	Hepatic venoocclusive disease and immune deficiency
	*TAF*1	X-linked Mental retardation 33
	*TDRD*7	Cataract 36
	*ZMYND*11	MRAD 30

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
