# Peer review of "Chromatin Structure and Dynamics: Focus on Neuronal Differentiation and Pathological Implication"

_genes, 2022, doi:10.3390/genes13040639_

Round 1

Reviewer 1 Report

The article entitled "Chromatin structure and dynamics: focus on neuronal differentiation and pathological implication" outlines the structure of chromatin, the mechanisms of gene expression regulated by epigenetic factors, and the importance of post-translational modifications (PTM) in neurodevelopment, especially neurogenesis and neural differentiation in an organized and informative way. This manuscript provides a comprehensive summary of the transcription mechanisms affected by DNA compaction, histone modification, as well as modification molecules. Additionally, the manuscript describes techniques for studying histone modification and summarizes disorders caused by abnormalities in histone modification mechanisms. Lastly, the authors discuss possible ways to modify histone modification abnormalities for clinical use. Thus, this review manuscript will appeal to a wide variety of readers, including students studying epigenetics, cell biologists and neuroscientists, as well as clinicians, and provide a profound understanding of histone modification in neurogenesis and neural differentiation. Nevertheless, some minor changes should be made.

  1. In the section on PTM during neuronal differentiation in chapter 5, detailed information is lacking. Especially in section 5.3, the authors summarized previous studies regarding the molecules involved in neuronal differentiation. There is, however, a lack of specific information, such as what brain regions and what types of cells are involved. As an example, the authors describe the NSCs in the Mll1 KO mice in lines 486-487. The details of the NSCs are not described. What tissue (the brain, neural tube, or another tissue) and what region (the cerebral cortex, hippocampus, or another region) do the NSCs reside in? Other parts have similar issues, such as EHMT2 in lines 505-507 and Sox19b in zebrafish in lines 515-517. These details should be provided.
  2. In section 5.1, the authors describe the neuronal differentiation process; however, it is unclear in what tissue and region the authors are describing neuronal differentiation. Please clarify.
  3. The authors describe the similarity of the epigenome between humans in vivo and organoids in vitro in lines 589-590 but go into greater detail about how they are similar.
  4. In lines 76-77, “Two forms of heterochromatin are…” appears twice in lines 78-79. Please remove one of them.
  5. In Table 1, HDAC10 is missing in histone deacetylase class II.
  6. In line 465, “H327ac” should be “H3K27ac.”

Author Response

Reply to Reviewer

We thank the reviewer for the reading and the corrections brought to improve our review. Please find here point by point the modifications made to the manuscript following your recommendations.

Comments to the Authors:

The article entitled "Chromatin structure and dynamics: focus on neuronal differentiation and pathological implication" outlines the structure of chromatin, the mechanisms of gene expression regulated by epigenetic factors, and the importance of post-translational modifications (PTM) in neurodevelopment, especially neurogenesis and neural differentiation in an organized and informative way. This manuscript provides a comprehensive summary of the transcription mechanisms affected by DNA compaction, histone modification, as well as modification molecules. Additionally, the manuscript describes techniques for studying histone modification and summarizes disorders caused by abnormalities in histone modification mechanisms. Lastly, the authors discuss possible ways to modify histone modification abnormalities for clinical use. Thus, this review manuscript will appeal to a wide variety of readers, including students studying epigenetics, cell biologists and neuroscientists, as well as clinicians, and provide a profound understanding of histone modification in neurogenesis and neural differentiation. Nevertheless, some minor changes should be made.

  1. In the section on PTM during neuronal differentiation in chapter 5, detailed information is lacking. Especially in section 5.3, the authors summarized previous studies regarding the molecules involved in neuronal differentiation. There is, however, a lack of specific information, such as what brain regions and what types of cells are involved. As an example, the authors describe the NSCs in the Mll1 KO mice in lines 486-487. The details of the NSCs are not described. What tissue (the brain, neural tube, or another tissue) and what region (the cerebral cortex, hippocampus, or another region) do the NSCs reside in? Other parts have similar issues, such as EHMT2 in lines 505-507 and Sox19b in zebrafish in lines 515-517. These details should be provided.

Response: We agree with the reviewer. We have clarified the origin of the cells:

  • In Section 5.3.1, we clarified the origin of rat NSCs.
  • hPSC is replaced by their specific cell line name “human ESC H9” in the VPA study and “HUES 9 ESC” in Qiao et al.
  • In the section on DOT1L, we mentioned that NPCs are derived from mESC
  • For the MLL1 studies, the definition of the mouse model has been clarified
  • For the MLL1 studies in zebrafish embryos, we specified that the study is on the development of the nervous system in zebrafish and that the population of NPC studied are SOX2-positive cells
  • For the study on EHMT2, we specified that the neuronal differentiation was realized from mouse neural crest derived cell lines (N2a cells)
  • For the study on Sox19b, we specified that the study is performed during zebrafish embryonic development and that NSCs are derived from the neural tube.

  1. In section 5.1, the authors describe the neuronal differentiation process; however, it is unclear in what tissue and region the authors are describing neuronal differentiation. Please clarify.

Response: We agree with the reviewer. We have clarified the neuronal differentiation described in this paragraph by specifying that it refers to the neuronal differentiation in the cerebral cortex and by specifying the location of radial glial cells in the ventricular zone and intermediate progenitors in the subventricular zone.

  1. The authors describe the similarity of the epigenome between humans in vivo and organoids in vitro in lines 589-590 but go into greater detail about how they are similar.

Response: We agree with the reviewer and have clarified this point by adding a paragraph that details the references mentioned highlighting the similarities between brain organoids development and fetal brain at the epigenetic and transcriptomic levels.

  1. In lines 76-77, “Two forms of heterochromatin are…” appears twice in lines 78-79. Please remove one of them.

Response: We remove the sentence written in duplicate

  1. In Table 1, HDAC10 is missing in histone deacetylase class II. 6. In line 465, “H327ac” should be “H3K27ac.”

Response: Thank you for this comment. HDAC10 has been added in Table 1 and H327ac has been replaced by H3K27ac.

Reviewer 2 Report

This review by Nothof et al. overviews the connection between epigenetic modifications and neuronal differentiation as well as the pathological consequences associated with deficits in the epigenetic machinery. The review was thoroughly written, starting from the basics of histone modification to deeper connections between epigenetics and neuronal development. The topic of this review is of great importance and interest, but I do have a few comments listed below. 

Major comments:

  1. My general impression after reading the manuscript is that though the authors overviewed a lot of details, it feels like a direct listing of a pile of information with only superficial explanations or understanding. For example, in line 410, section 5.3 when talking about the mechanisms of NSCs differentiation, the discussion is rather limited. For example, how the changes in chromatin will gradually limit access to multipotency genes, is it through the histone modification regulated chromatin structure or something else? Clearly, the discussions can be expanded to provide more insights and I suggest the authors go through the entire manuscript and deliberate in other places.
  2. On page 13, line 463 section 5.3.2, can the authors elaborate more on this observation: the differentiation is accompanied by deacetylation associated with a more compact chromatin while H3K79me2 is increased. If I understand correctly, H3K79me is the histone mark for the gene body. Does that mean a more compact chromatin would lead to differentiation with more gene expression? That doesn’t seem to align with the fact that the euchromatin, which is active in gene activity, is more open in terms of structure. Please include more discussion on this.
  3. When talking about the role of the H3K27me3 histone mark, I suggest the authors check out a relevant paper [PMID: 32841603]. In this paper, they discovered an intermediate compartment type (C) other than the traditional A/B compartment, which corresponds to the histone mark of H3K27me3, and it was found that the C compartment can have a high potential of conversion to the other two compartments under different circumstances. This conversion has some inner correlations with cell state development and tumorigenesis.
  4. And at the very end, I think it would be helpful to have more discussions on the role of the 3D architecture of chromatin. For example, [PMID: 31168090
  5. ] proposed that the heterochromatin serves as the driving force of compartmentalization for inverted and conventional nuclei in the retinal neurons. [PMID: 29365171, PMID: 31181064] quantitatively connect the 3D chromatin structure and histone modifications. [PMID: 26951677, PMID: 33086041] reconstructed the 3D organization of the whole-nucleus genome from basic histone information. After reading the manuscript, I feel that the 3D structure might be an interesting point to bring up and emphasize more, to connect the current illustration of the relationship between histone modifications and neuronal differentiation, from a more mechanistic perspective. I acknowledge a complete mechanistic connection might not be fully clear yet but maybe more deliberated discussions over the above point by citing these mentioned papers would be helpful in providing the community a supplementary thinking pipeline of mechanistically how epigenetic modification connects with neuronal differentiation.

Minor comments:

  1. The texting should be double-checked. For example, there should be a comma between “differentiation” and “Studying” in the abstract on line 16. The authors should check the grammar in other places across the manuscript.
  2. The authors need to make sure the figures are made by themselves otherwise the figures need to be properly cited.
  3. The quality of figures is low, it can be improved.

Author Response

Reply to Reviewer

We thank the reviewer for the reading and the corrections brought to improve our review. Please find here point by point the modifications made to the manuscript following your recommendations.

Suggestions for Authors

This review by Nothof et al. overviews the connection between epigenetic modifications and neuronal differentiation as well as the pathological consequences associated with deficits in the epigenetic machinery. The review was thoroughly written, starting from the basics of histone modification to deeper connections between epigenetics and neuronal development. The topic of this review is of great importance and interest, but I do have a few comments listed below.

Major comments:

  1. My general impression after reading the manuscript is that though the authors overviewed a lot of details, it feels like a direct listing of a pile of information with only superficial explanations or understanding. For example, in line 410, section 5.3 when talking about the mechanisms of NSCs differentiation, the discussion is rather limited. For example, how the changes in chromatin will gradually limit access to multipotency genes, is it through the histone modification regulated chromatin structure or something else? Clearly, the discussions can be expanded to provide more insights and I suggest the authors go through the entire manuscript and deliberate in other places.

Response: We agree with the reviewer.  We have completed the manuscript with additional information, notably on chromatin structure, DNA compaction, repressive marks, methylation and demethylation or 3D genome architecture (see below).

Moreover, we added a paragraph in the section 5.3.2 illustrating the loss of pluripotency of mESCs differentiating into NPCs with the associated deacetylation and histone marks for specific genes during neuronal differentiation. 

Also, for the H3K27me3 PTM, we implemented the discussion by adding data obtained in the zebrafish model.

  1. On page 13, line 463 section 5.3.2, can the authors elaborate more on this observation: the differentiation is accompanied by deacetylation associated with a more compact chromatin while H3K79me2 is increased. If I understand correctly, H3K79me is the histone mark for the gene body. Does that mean a more compact chromatin would lead to differentiation with more gene expression? That doesn’t seem to align with the fact that the euchromatin, which is active in gene activity, is more open in terms of structure. Please include more discussion on this.

Response: We thanks the reviewer for this comment. We added a paragraph in the 5.3.2 section. Observations made during in vitro neuronal differentiation of murine embryonic stem cells into neural progenitors show opposing global changes between the two histone PTMs: H3K79me2 and H3K27ac.

The combination of an overall gain of H3K79me2 at genes critical for neuronal development and a decrease of H3K27ac resulting in a more compact chromatin compared to mESCs allows for a specification towards the neuronal lineage.

  1. When talking about the role of the H3K27me3 histone mark, I suggest the authors check out a relevant paper [PMID: 32841603]. In this paper, they discovered an intermediate compartment type (C) other than the traditional A/B compartment, which corresponds to the histone mark of H3K27me3, and it was found that the C compartment can have a high potential of conversion to the other two compartments under different circumstances. This conversion has some inner correlations with cell state development and tumorigenesis.

We thank the reviewer for this observation. Indeed, information concerning the 3D compartmentalization, as well as information on the epigenetic marks associated with the different compartments has been added in the 2.1 part on chromatin structure. Moreover, in part 3.3.1 concerning the repressive marks, in the paper mentioned by reviewer 2, an increase of the H3K27me3 mark in compartment I associated with an increase of the H3K9me3 mark in compartment B is observed at the level of downregulated genes in colon cancer. The mechanisms of block hypomethylation found in B and I compartments are characteristic of   excessively divided cells. The recovered intermediate compartment is named I in the mentioned article and not C.

  1. And at the very end, I think it would be helpful to have more discussions on the role of the 3D architecture of chromatin. For example, [PMID: 31168090 5. ] proposed that the heterochromatin serves as the driving force of compartmentalization for inverted and conventional nuclei in the retinal neurons. [PMID: 29365171, PMID: 31181064] quantitatively connect the 3D chromatin structure and histone modifications. [PMID: 26951677, PMID: 33086041] reconstructed the 3D organization of the whole-nucleus genome from basic histone information. After reading the manuscript, I feel that the 3D structure might be an interesting point to bring up and emphasize more, to connect the current illustration of the relationship between histone modifications and neuronal differentiation, from a more mechanistic perspective. I acknowledge a complete mechanistic connection might not be fully clear yet but maybe more deliberated discussions over the above point by citing these mentioned papers would be helpful in providing the community a supplementary thinking pipeline of mechanistically how epigenetic modification connects with neuronal differentiation.

Response: We thank the reviewer for this remark. To address this comment, we introduced in section 2.1 the notions of A and B compartmentalization as well as the notion of an inverted core. We have used this article to define the importance of heterochromatin in the A and B compartimentalization in both conventional and inverted nuclei and the importance of lamina-heterochromatin interactions in establishing the conventional nucleus. To complete the description of the 3D organization of the genome, we have defined the notion of TAD in section 2.2.

A short chapter has been added in the neuronal differentiation section on the 3D modifications of the genome during neuronal differentiation (part 5.3.3). We also discussed the predictive tools available for the three-dimensional study of the genome citing the articles mentioned in reviewer 2’s comment. Furthermore, this review focuses mainly on histone modifications during neuronal differentiation. The study of the link between 3D compartmentalization and neuronal differentiation is relevant, but is already reviewed in recent reviews by Kishi et al., Arzate-Bejía et al. and Ghosh et al.

Minor comments:

  1. The texting should be double-checked. For example, there should be a comma between “differentiation” and “Studying” in the abstract on line 16. The authors should check the grammar in other places across the manuscript.

Response: We agree with the reviewer. We have corrected some mistakes in the manuscript.

  1. The authors need to make sure the figures are made by themselves otherwise the figures need to be properly cited.

Response: All figures were made by the author using BioRender. It is mentioned at the end of the manuscript in the acknowledgments.

  1. The quality of figures is low, it can be improved

Response: In response to this comment, we have replaced the figures by uploading them directly to BioRender with a high-resolution quality. Regarding the global quality of figures, the goal was to illustrate in a simplistic manner the involvement of chromatin structure and dynamics, especially during neuronal differenciation adressing students, neuroscientists, as well as clinicians.

Round 2

Reviewer 2 Report

The authors have addressed most of my concerns and the revised manuscript is greatly improved. I am particularly impressed at the effort they made in including the perspective of the 3D genome and I am now recommending the manuscript to be accepted.